# Swiss-Chinese Cooperation in Tropical Medicine: The Role of Professor Marcel Tanner

**DOI:** 10.3390/diseases10040083

**Published:** 2022-10-08

**Authors:** Shan Lv, Wei Ding, Robert Bergquist, Guojing Yang, Jiagang Guo, Xiao-Nong Zhou

**Affiliations:** 1National Institute of Parasitic Diseases, Chinese Center for Diseases Control and Prevention, Shanghai 200025, China; 2NHC Key Laboratory of Parasite and Vector Biology, WHO Collaborating Centre for Tropical Diseases, National Center for International Research on Tropical Diseases, Ministry of Science and Technology, Shanghai 200025, China; 3School of Global Health, Chinese Center for Tropical Diseases Research, Shanghai Jiao Tong University School of Medicine, Shanghai 200025, China; 4Ingerod, SE-454 94 Brastad, Sweden; 5School of Tropical Medicine, Hainan Medical College, Haikou 571199, China; 6Department of Neglected Tropical Diseases, World Health Organization, 1211 Geneva, Switzerland

**Keywords:** schistosomiasis, China, Switzerland, cooperation, tropical medicine, tribendimidine

## Abstract

This paper is in honour of Professor Marcel Tanner, President of the Swiss Academies of Arts and Sciences, and former Director of the Swiss Tropical and Public Health Institute (Swiss TPH), in Basel, Switzerland. In the 30 plus years since his first visit to China in 1989, Professor Tanner has tirelessly promoted research collaboration between Switzerland and China on health and tropical diseases through international meetings, scholar exchange, and training of young scientists. As a contribution to Professor Tanner’s life’s work of collaboration with Chinese scientists, we summarize here ideas conceived, work initiated and major outcomes. His approach, embodied in his flowery expression: “Alps and Himalayas never meet, but Swiss and Chinese can”, marked the occasion in 2013 when Xinhua Co., Ltd., a pharmaceutical company in Shandong of China, agreed to produce tribendimidine, a new remedy for tropical helminth infections, that was the fruit of long-term research by scientists at the Swiss TPH in Basel, and National Institute of Parasitic Diseases (NIPD) in Shanghai. This was neither the first nor the last of Professor Tanner’s forceful, yet diplomatic influence, and we follow in his footprints when continuing in Swiss-Chinese cooperation in tropical medicine.

## 1. The World Bank Loan for Schistosomiasis Control

Professor Tanner’s interest in China and his strong commitment to build a sustained collaboration with Chinese scientists coincided with the opening up of the country in the 1980s. His first visit in 1989 soon led to a strong involvement, assisted by his work for the 10-year World Bank Loan Project (WBLP) on schistosomiasis control in China, initiated in 1992 to further improve China’s previous spectacular achievements in schistosomiasis control. According to Mao and Shao [1], the main drivers in this context were an early recognition of the problem, sustained commitment, intersectoral collaboration, community participation and political will. The positive outcome of clinical trials of the new drug praziquantel [2] and its application for mass drug administration (MDA) helped to turn the situation around, and led to the WBLP for controlling schistosomiasis in China [3]. This intervention was active in the 1992–2001 period and enabled further significant progress by adapting MDA to the Chinese situation of endemicity, which differs in many aspects from that in Africa and other parts of the world. Of note, a large number of mammalian hosts are naturally infected with *Schistosoma japonicum* (the only species of human schistosome in China) and this parasite’s intermediate snail host, *Oncomelania hupensis*, is amphibious.

Before the implementation of the WBLP, an integrated strategy with emphasis on snail control had been implemented in China for decades. The arrival of praziquantel in the early 1980s caused the World Health Organization (WHO) to change its recommendation to MDA as the core approach; the strategy, therefore, used in the WBLP approach. However, the previous success of snail control against schistosomiasis made Chinese health managers and scientists feel that this tool should not be abandoned, but remain an important part of the overall approach. Therefore, the Chinese Government provided supplementary financial resources to support continued snail control.

## 2. The Joint Research Management Committee

Since operational research was considered an important component of the WBLP, the Joint Research Management Committee (JRMC) was established as an operational research arm. It was given a small part of the WBLP budget to stimulate needed research in the eight endemic provinces. The UNDP/World Bank/WHO Special Programme for Research and Training in Tropical Diseases (TDR) sponsored the inclusion of four international experts to work together with eight Chinese counterparts under the chairmanship of Dr Huanzeng Wang sponsored by the Chinese Ministry of Health (Figure 1). It can be argued that the most important legacy of the WBLP might not have been the immediate strong reduction in transmission and number of infected cases, but future new drugs, improved diagnostics and vaccine development delivered by the JRMC.

As one of the international members of the JRMC, Professor Tanner visited many schistosomiasis endemic areas and institutes/centers for schistosomiasis control (Figure 2). Early on, he helped researchers to write a scientific proposal to WHO/TDR that was eventually funded. The Hunan Institute of Schistosomiasis Control (HISC) in Yueyang was the site of the first meeting of the JRMC, and a close relationship was established between Professor Tanner and the HISC researchers that included Dr. Yuesheng Li, who would later be promoted to be Director of HISC. Prior to WBLP, Professor Tanner had organized the application of making HISC a WHO Collaborating Centre on Schistosomiasis Control in Lake Regions, which was approved in May, 1990. Professor Tanner was also engaged in joint technical and personal trainings via workshops and personal exchange between HISC and the Swiss Tropical and Public Health Institute (Swiss TPH) and was pleased to attend the celebration of the 60th anniversary of HISC in 2018.

During his time at the JRMC, Professor Tanner witnessed the development of novel technologies in China, which provided a robust force in the fight against schistosomiasis, resulting in the later significant decrease in its prevalence. He convinced the JRMC of the need to incur a broad research agenda and realized the need for a drug development project when Professor Shu-Hua Xiao at the National Institute of Parasitic Diseases (NIPD) in Shanghai, China proposed investigating artemether, a promising new compound with anti-schistosomal properties. Interestingly, artemether had originally been developed as a new anti-malaria drug by Professor Youyou Tu [4], and was subsequently shown to also be active against *Schistosoma* spp., a research approach that was financially supported by JRMC.

## 3. Swiss-Chinese Cooperation in Tropical Medicine

After the WBLP, Swiss TPH colleagues extended cooperation with Chinese researchers, and Professor Tanner supported Swiss projects to be conducted in China moving forward on application and implementation of new joint research projects. Experts from different institutes jointly carried out several projects on health care, drug development (artemether, 1,2,4-trioxolanes and tribendimidine), capacity building, and information dissemination [5,6]. With time, the scope moved from single subjects or diseases to broadened approaches, including global health issues. This research has played an important role not only in China, but also for global health cooperation with reference to endemic countries, especially in Africa and Southeast Asia, and will no doubt in the future make profound contributions to the prevention and treatment of tropical diseases. 

The work on artemether and derivatives initiated when JRMC was active led to a close, long-term collaboration between Professor Tanner and Professor Shu-Hua Xiao that eventually resulted in broad drug development including work on compounds also active on tropical, endemic diseases other than schistosomiasis. To that end, Professor Xiao regularly spent several months each year at the in Swiss TPH in the period from 1998 to 2008 (Figure 3). During this period, much research was carried out, first on artemether and later also on other compounds with anthelmintic activity, eventually resulting in the publication of 26 articles in peer-reviewed journals of high standing. Important contributions included work on the level of action against *S.*
*japonicum* life stages [7], on various species [8], on the possibility of inducing immunity against this worm by using artemether to block parasite maturation [9] and on tribendimidine, a novel drug with effect against *Ankylostoma*, *Ascaris* and *Enterobius*, as well as *Clonorchis* and *Opisthorchis* [10]. Successful clinical trials [11] carried out under the auspices of the NIPD-Swiss TPH cooperation led to the commercial production of tribendimidine by Xinhua Co., Ltd. in Shandong of China.

Professor Tanner also played an important role for improving the quality of malaria elimination efforts in China, paying two visits to Hainan Province, once an area hyper-endemic for malaria with high year-round transmission. He was instrumental in assisting Chinese malaria experts in their work to free China of malaria transmission, efforts that resulted in China’s certification as malaria-free by the WHO on 30 June 2021 [12]. The fact that the national malaria program and its partners could eventually bring 30 million cases of malaria down to zero indigenous cases with only a manageable number of imported cases was nothing less than a remarkable achievement.

## 4. Training and Exchange

Capacity building was strengthened through short training courses, Masters, PhD and Postdoctoral training. For instance, Drs. Jiagang Guo, Guojing Yang, and Fuqiang Cui were supervised by Professor Tanner, followed by Dr. Shan Lv, who was supervised by Professor Jürg Utzinger, and Dr. Yingsi Lai by Professor Penelope Vounatsou. They all successfully defended their PhD theses at the Swiss TPH and now play active roles in disease control and research in China. In addition, many young Chinese scientists have attended training courses organized by Professor Tanner, on malaria and other tropical diseases, with a strong focus on management, such as priority setting, resource allocation and strategic planning. In 2015, Ms. Bernadette Peterhans and Dr. Axel Hoffmann of the Swiss TPH hosted a two-week workshop on program/project planning and management, with a special focus on health care and disease control at the NIPD together with Drs. Peiling Yap and Wei Ding in Shanghai, with 14 participants from the NIPD and provincial disease control centers in China. Importantly, several PhD and Master students from the Swiss TPH have pursued their projects in China since 2005. Dr. Peter Steinmann and Dr. Peiling Yap from the Swiss TPH carried out their postdoctoral research in China under supervision by Professor Xiao-Nong Zhou, Director of the NIPD.

Involvement by young scientists in internal projects opened the door to pursue Masters and Doctoral degrees as well as to apply for Postdoctoral training overseas. One such young scientist, Dr. Jiagang Guo from NIPD, was the initial and foremost JRMC secretary. Benefitting from the WBLP experience, he later became the first Chinese PhD student in the Swiss TPH, where he was supervised by Professor Tanner. After having successfully defended his PhD thesis in Switzerland, Dr. Guo led the Department of Schistosomiasis Control at NIPD for more than 10 years and was deeply involved in the design and implementation of the national program of schistosomiasis control. In 2012, he was appointed as civil servant in the Department of Neglected Tropical Diseases at the WHO’s headquarters in Geneva, Switzerland.

## 5. Closing the Information Gap

Discussing the importance of collaboration and controlling the endemic NTDs, a group of 20 conditions that are mainly endemic in tropical mostly impoverished areas in Southeast Asia, Professors Tanner and Zhou realized the importance of rapidly releasing recent results in top-level journals. *Acta Tropica* was first approached, and agreed to publish submitted papers after peer-review in 2010. This led to a series of special issues, alternatively in *Advances in Parasitology* and *Acta Tropica*, starting in 2010 (Table 1). In addition, several high impact articles were published in *The Lancet* and *The Lancet Infectious Diseases* [13,14] (Table 2). 

Thanks to growing connections with scientists abroad, Chinese scholars now increasingly publish in international journals. Joint research covering parasitic diseases, such as malaria, schistosomiasis, clonorchiasis and soil-transmitted helminthiases, by NIPD and the Swiss TPH, resulted in more than 100 publications in peer-reviewed journals. Activities have not only paid attention to the control of parasitic diseases in China, but have also introduced Chinese researchers to work in other endemic countries.

## 6. NTD Elimination in China

The verification by the WHO of the elimination of lymphatic filariasis in China in 2007 [15] triggered an elimination process of major parasitic diseases in the country, with sights first set on malaria and schistosomiasis. The former was achieved in 2021 and the latter will be in 10 years. However, it was understood that sensitive surveillance and response systems would urgently be needed for domestic elimination of tropical diseases. Feeling that the elimination experiences in China could be shared with the global scientific community, Professors Tanner and Zhou proposed an international forum on Surveillance-Response System (SRS) to provide a platform for information exchange on research in tropical disease. The SRS forum grew in importance and became a biannual international conference co-sponsored by the NIPD, Swiss TPH and WHO, with the aim of updating research and activities seeking both national and global solutions to address the challenges ahead. The SRS meetings brought together stakeholders such as scientists, project managers, decision makers, donors and manufacturers, with the first held in Shanghai in June 2012 under the title “SRS Leading to Tropical Diseases Elimination” (Figure 4). The sixth forum was held in June 2022, but had to be organized as a virtual event due to the COVID-19 pandemic.

The SRS forums have attracted a growing number of participants and have yielded highly fruitful results, e.g., demonstration projects on malaria/schistosomiasis control and elimination in Tanzania and Zimbabwe [16] and a network on echinococcosis and cysticercosis, supported by the Belt and Road Global Infrastructure Development Strategy adopted by the Chinese Government in 2013 [17]. The SRS meetings have underpinned the collaboration among Chinese institutions with global institutions by providing a platform for exchanging academic findings, sharing fieldwork experiences and debating solutions. They have had a far-reaching impact on tropical disease research, allowing NIPD and its Chinese partners to gain unique opportunities for cooperation with African and Southeast Asian countries, e.g., in the form of the Institution-based Network on China–Africa Cooperation for Schistosomiasis Elimination (INCAS) and the Institution-based Network on China-Africa Cooperation for Malaria Elimination (INCAM) established in 2015 and 2019, respectively [17,18].

In 2012, Professor Zhou initiated the international journal *Infectious Diseases of Poverty* (IDoP), which was launched with the concept of “One Health, One World” [19]. With Professor Zhou as Editor-in-Chief and Professor Tanner on the Editorial Board, the journal has had a strong global impact through its thematic series and articles on essential topics, prioritizing research and field application for One Health, an integrated and unifying approach to sustainably balance and optimize the health of people, animals and ecosystems. During 2012–2015, Professor Tanner contributed by assisting the scientific quality of IDoP by submitting nine papers covering a broad area from NTD elimination to EcoHealth research, the publication of which accounted for 6% of the publication record at that time. These papers were not only examples of science linked with practical implementation, but also inaugurated the role of One Health.

Ten years later Professor Zhou, now also serving as Vice Dean of School of Global Health, Chinese Center for Tropical Diseases Research, Shanghai Jiao Tong University School of Medicine, with help from Professor Tanner as co-Editor-in-Chief, established *Science in One Health* (https://www.journals.elsevier.com/science-in-one-health (accessed on 20 August 2022)), a new journal in the One Health sphere which aims to promote more scientific advances in One Health research, following the WHO’s guidelines for the disease control and elimination through EcoHealth approaches, including nutrition and environmental health. In 2020, Shanghai Jiao Tong University and University of Edinburgh co-established the One Health Center, with Professor Zhou as Director and Professor Utzinger as Member of Science and Technology Advisory Committee. Professor Jakob Zinsstag, a leading One Health scientist at the Swiss TPH, currently assists by organizing bilateral cooperation, including symposia and projects.

## 7. International Outreach

In 1957, Swiss TPH founded the Ifakara Health Institute (IHI) in Tanzania. As a long-term partner of Swiss TPH, it has developed into one of the most renowned research institutions in Africa. When NIPD was seeking its first overseas public health project partner in 2013, Professor Tanner instigated a successfully collaboration between NIPD with IHI. The two institutes jointly initiated the tripartite cooperation project China-Tanzania Pilot Project of Malaria Control: Application of Community-based and Integrated Strategy, supported by a grant from the British Department for International Development in 2015 under the China-United Kingdom Global Health Support Programme (http://cps.nhc.gov.cn/ghspen/index.shtml (accessed on 20 August 2022)). Professor Zhou took the leadership, as the principle investigator of the project, to adapt China’s domestic experience tailored to local context by applying the 1,7-malaria Reactive Community-based Testing and Response Strategy (1,7-mRCTR), which reduced the malaria burden by 81% in the intervention area [20]. In 2019, the Bill and Melinda Gates Foundation found this outcome to be of such importance that they offered financial support for validation and scaling up activities. Since 2020, this study has been sustained by the National Health Commission of China.

In 2011, an international project, with Professor Zhou as the principal investigator, was granted financial support from the International Development and Research Center of Canada (IDRC). This was administered under the framework of the Regional Network of Asian Schistosomiasis and Other Important Helminth Zoonoses (RNAS+) [21], and aimed to design socio-ecosystem strategies to effectively control emerging helminth zoonoses in Southeast Asia. The project, carried out in Cambodia, China, Laos, the Philippines, Thailand and Vietnam, marked the first time Chinese researchers led and implemented a multi-country project on tropical diseases. Since 2011, Professors Jürg Utzinger and Peter Odermatt at the Swiss TPH contributed strongly to forward and conclude this project.

## 8. The Future

Although Professor Tanner retired as Director of the Swiss TPH many years ago, he remains devoted to global health and still pays close attention to the progress of research and control with respect to tropical diseases—well summed up in his own words: “Retired but not tired”. Professor Utzinger, his successor at Swiss TPH, is continually involved in the Swiss-Chinese cooperation initiated by Professor Tanner, who in 2021 was awarded the Friendship Award by the Chinese Government for his contribution to tropical medicine (Figure 5).

In his meeting with the delegation from Xinhua Co., Ltd. in January 2013, Professor Tanner used the old adage: “if you want to go fast, walk alone; if you want to go far, walk together”. We believe that this is a good summary of how the successes were reached over the last 30 plus years, through collaboration over borders. Not only does it ensure that there is a road ahead, it also underlines that the collaboration initiated by Professor Tanner and his Chinese colleagues can continue, and indeed be strengthened, by the young generation now coming of age.

## Figures and Tables

**Figure 1 diseases-10-00083-f001:**
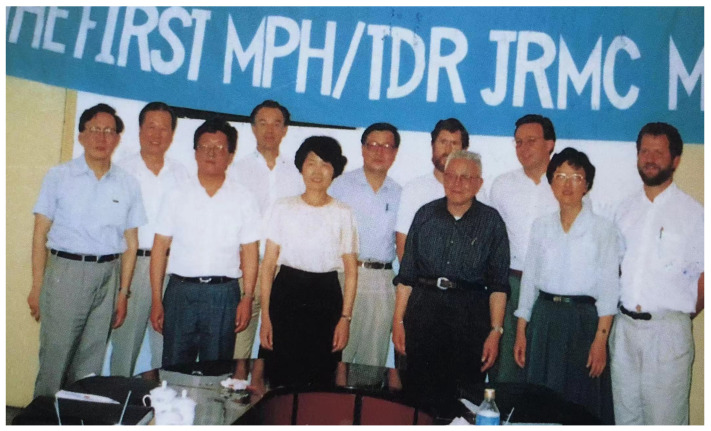
The first board meeting of the Joint Research Management Committee in Yueyang, China in 1992. From left to right: Drs. Minggang Chen, Hongchang Yuan, Jiang Zheng, Robert Bergquist, Qingsi Zheng, Huanzeng Wang, David Evans, Ke Qian, Marcel Tanner, Jie Chen, Bruno Gryseels.

**Figure 2 diseases-10-00083-f002:**
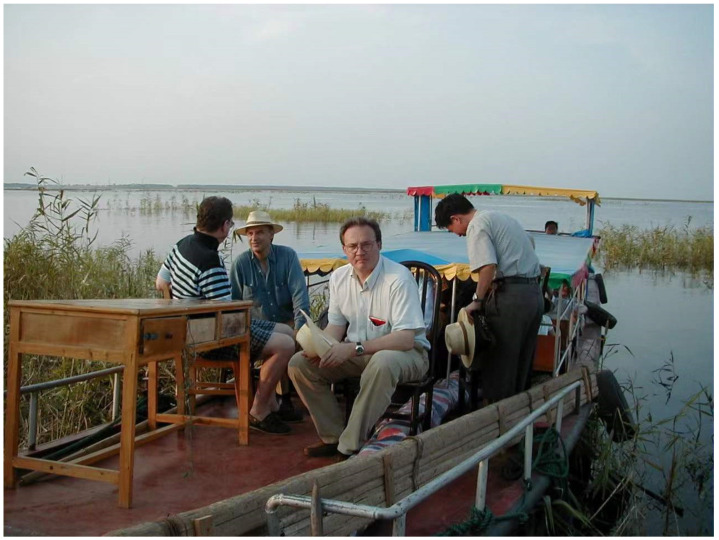
Professors Tanner (middle right) and Bergquist (middle left) on a tour by boat along the Dongting Lake, a major schistosomiasis endemic area, in 2001.

**Figure 3 diseases-10-00083-f003:**
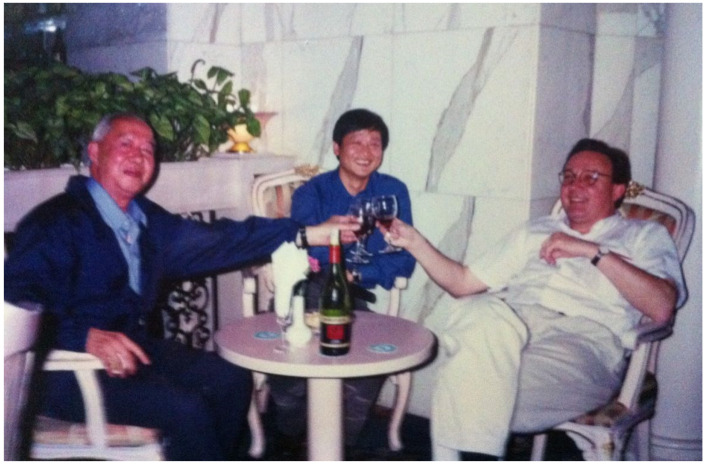
Professor Tanner (right) together with his Chinese friends Professor Shu-Hua Xiao (left) and Dr. Jiagang Guo (middle) in Shanghai in 2001.

**Figure 4 diseases-10-00083-f004:**
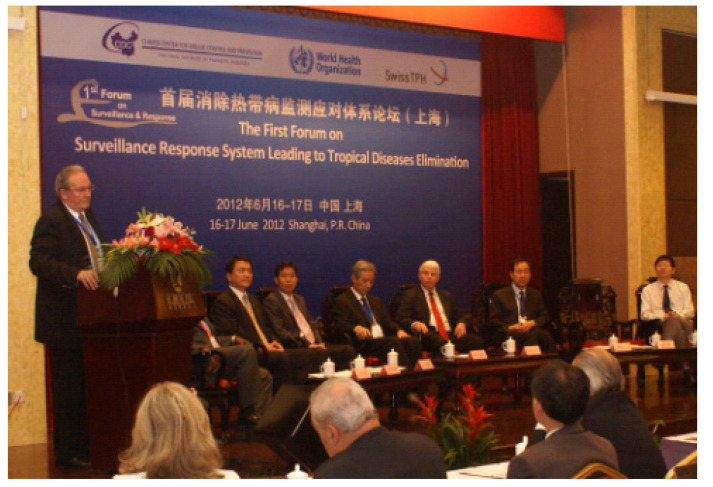
Professor Tanner, giving the opening speech as Chairman of the First Forum on Surveillance and Response System Leading to Tropical Diseases Elimination in Shanghai 2012.

**Figure 5 diseases-10-00083-f005:**
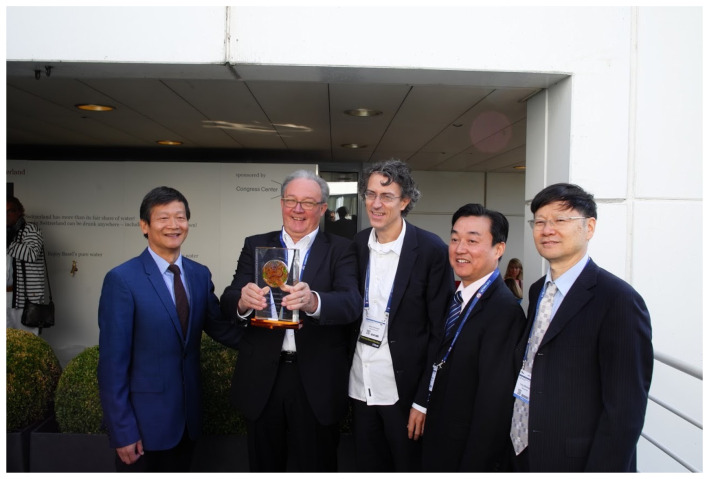
Professor Tanner with the award of *Excellent Advisor* to celebrate his 20th anniversary of cooperation between NIPD and Swiss TPH, presented by Professor Zhou on behalf of NIPD at the Swiss TPH in Basel, Switzerland, in 2015. From left: Dr Jiagang Guo, Professor Marcel Tanner, Professor Jürg Utzinger, Professor Xiao-Nong Zhou, Professor Wei-Zhong Yang.

**Table 1 diseases-10-00083-t001:** Special Issues of journals dealing with tropical diseases published through international cooperation.

Topic	Guest Editors	Journal	Year
Important helminth infections in Southeast Asia: diversity, potential for control and prospects for elimination	Utzinger J., Bergquist R.,Olveda R., Zhou X.N.	*Advances in* *Parasitology*	2010
Control and elimination of helminth infections in Asia	Utzinger J., Brattig N.W., Leonardo L., Zhou X.N., Bergquist R.	*Acta Tropica*	2015
The role of RNAS * for networking and global engagement in work on tropical diseases: milestones achieved	Leonardo L., Bergquist R., Utzinger J., Willingham A.L., Olveda R., Zhou X.N.	*Advances in* *Parasitology*	2019
Helminthiases in the People’s Republic of China: Status and prospects	Brattig N.W., Bergquist R., Qian M.B., Zhou X.N., Utzinger J.	*Acta Tropica*	2020
A malaria-free China: global importance and key experiences	Yin J.H., Lengeler C., Tanner M., Zhou X.N.	*Advances in* *Parasitology*	2022

* Regional Network on Asian Schistosomiasis and other Helminthic Zoonoses.

**Table 2 diseases-10-00083-t002:** High-impact articles on parasitic diseases resulting from collaborative research between NIPD and Swiss TPH.

Title	Authors	Journal	Year
Schistosomiasis control: experiences and lessons from China	Wang L., Utzinger J., Zhou X.	*The Lancet*	2008
China’s sustained drive to eliminate neglected tropical diseases	Yang G.J., Liu L., Zhu H.R., Griffiths S.M., Tanner M., Bergquist R., Utzinger J., Zhou X.N.	*The Lancet Infectious Diseases*	2014
Clonorchiasis	Qian M.B., Utzinger J., Keiser J., Zhou X.N.	*The Lancet*	2016
Enhancing collaboration between China and African countries for schistosomiasis control	Xu J., Yu Q., Tchuenté L.A., Bergquist R., Sacko M., Utzinger J., Lin D.D., Yang K., Zhang L.J., Wang Q., Li S.Z., Guo J.G., Zhou X.N.	*Lancet Infectious Diseases*	2016
Neglected tropical diseases in the People’s Republic of China: progress towards elimination	Qian M.B., Chen J., Bergquist R., Li Z.J., Li S.Z., Xiao N., Utzinger J., Zhou X.N.	*Infectious Diseases of Poverty*	2019

## Data Availability

Not applicable.

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
