# Peer review of "Swiss-Chinese Cooperation in Tropical Medicine: The Role of Professor Marcel Tanner"

_diseases, 2022, doi:10.3390/diseases10040083_

Round 1
Reviewer 1 Report
I really liked the presented manuscript. In my view, such "reminiscence" manuscripts are very illuminative.
1) Is the first author's last name is LV?
2) Please unify Journal names spelling (some with periods, some without, etc.)
3) Please make sure references in foreign language, f.e. 5 define the language of the cited reference (in Chinese). If I understand it right, the Zhongguo Yao Li Xue Bao is Acta Pharmacologica Sinica.
4) Please unify titles in the References - word capitalization, Italics.
Author Response
Point #1 Is the first author's last name is LV?
Response: Many thanks for the reminding. The last name “LV” is correct. According to the reviewer 2 (see point #14 below), the name is changed to lowercase, i.e. Lv. (see the revised manuscript).
Point #2 Please unify Journal names spelling (some with periods, some without, etc.)
Response: We have change the reference style according to the recent publications in the journal and unified the Journal name spelling.
Point #3 Please make sure references in foreign language, f.e. 5 define the language of the cited reference (in Chinese). If I understand it right, the Zhongguo Yao Li Xue Bao is Acta Pharmacologica Sinica.
Response: We had changed the journal name to “Acta Pharmacol. Sin.” (see the references)
Point #4 Please unify titles in the References - word capitalization, Italics.
Response: we have unified the titles in reference throughout.

Reviewer 2 Report
This manuscript nicely summarises the long-standing collaboration between Swiss and Chinese institutions to control parasite diseases, mainly in China. The collaboration led by Professor Marcel Tanner succeeded in controlling schistosomiasis, educated many students and was prolific.
Moreover, I have some specific comments:
Lines 46-47: “which differs in many aspects from that in Africa and other parts of the world”. Could you explain what makes the Chinese situation different?
Lines 52-55: It is not obvious to me what you mean with this sentence. Do you mean that the WBLP was pioneering?
Line 63: could you list the eight endemic provinces?
Line 63: what is UNDP?
Line 92: “Schistosoma” should be “Schistosoma spp” and in italics.
Line 97-100: could you give a reference?
Line 114: do you mean Schistosoma japonicum or Schistosoma spp? Please, indicate one of both.
Lines 116-117: Please, use italics for genera.
Line 132: The author “Shan LV” is in capital letters. However, the authors here refer to “Shan Lv” (lowercase v).
Line 140: You use “Drs” here. Whereas, you use “Drs.” in other sentences. Please, unify.
Line 159: Please, define NTD and NTDs.
Line 206: you could define “one health” concept.
Line 212-213: Please, add a verb in this sentence. “These papers ARE not only examples of …”.
Line 226: “the Swiss founded…” who is the Swiss? Is Prof Tanner? Or do you mean Switzerland?
Lines 242-243 : Do you mean Canada?
Line 248: “Professors Jurg Utzinger”, I think it should be “Professors Jürg Utzinger”
Line 253: I think you do not need “but”.
Final suggestion: the authors might want to acknowledge the staff involved in the collaboration over the last decades but are not listed as authors of this manuscript.
Author Response
Point #5 This manuscript nicely summarises the long-standing collaboration between Swiss and Chinese institutions to control parasite diseases, mainly in China. The collaboration led by Professor Marcel Tanner succeeded in controlling schistosomiasis, educated many students and was prolific.
Response: Many thanks for the positive comments.
Point #6 Lines 46-47: “which differs in many aspects from that in Africa and other parts of the world”. Could you explain what makes the Chinese situation different?
Response: We think the difference between China and other schito-endemic countries, particularly in Africa lies in the transmission network, including mammalian reservoirs and the intermediate host. The further explanation is the following sentence, i.e., “Of note, large number of mammalian hosts are naturally infected with Schistosoma japonicum (the only species in China) and this parasite’s intermediate snail host, Oncomelania hupensis, is amphibious”
Point #7 Lines 52-55: It is not obvious to me what you mean with this sentence. Do you mean that the WBLP was pioneering?
Response: We revised this sentence. We mean that WHO changed the recommendation to MDA for schistosomiasis control after the arrival of praziquantel. However, Chinese scientists agree with MDA in WBLP, but still persisted the snail control, which invested by Chinese government.
Point #8 Line 63: could you list the eight endemic provinces?
Response: We added the name of the eight provinces (see the revised manuscript).
Point #9 Line 63: what is UNDP?
Response: we added the full name of UNDP in the text, i.e., the United Nations Development Programme.
Point #10 Line 92: “Schistosoma” should be “Schistosoma spp” and in italics.
Response: we revised the name accordingly and wrote it in italics.
Point #11 Line 97-100: could you give a reference?
Response: we provided two references showing the joint projects between Swiss TPH and NIPD.
Point #12 Line 114: do you mean Schistosoma japonicum or Schistosoma spp? Please, indicate one of both.
Response: according to the reference, the species should be Schistosoma japonicum. We had specified the species in the revised manuscript.
Point #13 Lines 116-117: Please, use italics for genera.
Response: the italic genera names have been provided in the revised manuscript.
Point #14 Line 132: The author “Shan LV” is in capital letters. However, the authors here refer to “Shan Lv” (lowercase v).
Response: Many thanks. We have already changed the name in the title page. Lowercase v is right.
Point #15 Line 140: You use “Drs” here. Whereas, you use “Drs.” in other sentences. Please, unify.
Response: We have use the “Drs.” throughout the text.
Point #16 Line 159: Please, define NTD and NTDs.
Response: we provided the definition of the term according to the World Health Organization.
Point #17 Line 206: you could define “one health” concept.
Response: we provided the recent definition proposed by One Health High Level Expert Panel in 2021.
Point #18 Line 212-213: Please, add a verb in this sentence. “These papers ARE not only examples of …”.
Response: we are grateful for the suggestion and had added the word “are” in the sentence.
Point #19 Line 226: “the Swiss founded…” who is the Swiss? Is Prof Tanner? Or do you mean Switzerland?
Response: this shall be “Swiss TPH”. According to the homepage of IHI, Dr. Rudolf Geigy from Swiss Tropical Institute in Basel established Swiss Tropical Institute Field Laboratory (STIFL) in Tanzania and later became IHI.
Point #20 Lines 242-243 : Do you mean Canada?
Response: We are sorry for this wrong spelling. This has been corrected in the revised manuscript.
Point #21 Line 248: “Professors Jurg Utzinger”, I think it should be “Professors Jürg Utzinger”
Response: We had revised the name accordingly.
Point #22 Line 253: I think you do not need “but”.
Response: This is good suggestion. We had removed the word.
Point #23 Final suggestion: the authors might want to acknowledge the staff involved in the collaboration over the last decades but are not listed as authors of this manuscript.
Response: We much appreciate the suggestion. The major contributors to the review and cooperation in tropical medicine between China and Swiss TPH had been provided (see the revised manuscript).
